# Factors affecting healthcare workers' compliance with social and behavioural infection control measures during emerging infectious disease outbreaks: rapid evidence review

Samantha K Brooks ![ORCID], N Greenberg, Simon Wessely ![ORCID], G J Rubin

Department of Psychological Medicine, King's College London, London, UK

**Correspondence to**
Dr Samantha K Brooks;
samantha.k.brooks@kcl.ac.uk

## ABSTRACT

**Objective** The 2019–2020 outbreak of novel coronavirus has raised concerns about nosocomial transmission. This review's aim was to explore the existing literature on emerging infectious disease outbreaks to identify factors associated with compliance with infection control measures among healthcare staff.

**Methods** A rapid evidence review for primary studies relevant to healthcare workers' compliance with infection control measures.

**Results** Fifty-six papers were reviewed. Staff working in emergency or intensive care settings or with contact with confirmed cases appeared more likely to comply with recommendations. There was some evidence that anxiety and concern about the risk of infection were more associated with compliance, and that monitoring from superiors could improve compliance. Observed non-compliance of colleagues could hinder compliance. Staff identified many barriers to compliance related to personal protective equipment, including availability, perceived difficulty and effectiveness, inconvenience, discomfort and a negative impact on patient care. There were many issues regarding the communication and ease of understanding of infection control guidance.

**Conclusion** We recommend provision of training and education tailored for different occupational roles within the healthcare setting, managerial staff 'leading by example', ensuring adequate resources for infection control and timely provision of practical evidence-based infection control guidelines.

## INTRODUCTION

The recent years have seen frequent outbreaks of emerging infectious diseases.[1] Examples include severe acute respiratory syndrome (SARS) in 2003, avian influenza (H5N1) in 2004, swine influenza (H1N1) in 2009, Middle East respiratory syndrome (MERS) in 2015 and most recently coronavirus SARS-CoV-2 originating in 2019. Previous outbreaks saw high levels of nosocomial transmission (hospital-acquired infections).[2]

### STRENGTHS AND LIMITATIONS OF THIS STUDY

⇒ This review synthesises existing literature on healthcare workers' compliance with infection control measures during infectious disease outbreaks, providing an overview of what has already been established and identifying gaps in the literature.

⇒ The search strategy was broad, and reference lists of included papers were hand-searched for any additional relevant papers which may have been missed.

⇒ The search was limited to English-language papers due to the rapid nature of the review.

⇒ Included papers did not undergo formal quality appraisal, again due to the rapid nature of the review.

A major cause of nosocomial transmission is poor compliance with personal protective behaviours among healthcare staff.[3 4] Early reports from the COVID-19 frontline have noted poor compliance of healthcare workers with recommended behaviours: in one hospital in China, many had their masks hung from one ear or pulled downwards, while more than half had inadequate hand hygiene.[5]

Compliance with infection control behaviours can be difficult. Previous literature has reported on difficulties in the general population with adhering to protective behaviours such as facemask wearing,[6] using hand sanitising gel[7] and quarantine[8] during infectious disease outbreaks. The main facilitators of compliance in the general population appear to be perceived susceptibility, perceived severity of being afflicted and perceived benefits of compliance, as well as accurate knowledge about the disease and the recommended behaviours, while major barriers include discomfort, embarrassment and practical issues.[6 8] A review[9] of healthcare workers' perceptions of barriers

and facilitators to compliance with guidelines during respiratory outbreaks suggested that protective practices are influenced by understanding of guidelines, support received from managers, communication about guidelines, sufficient resources, perceived value of following guidance, comfort of personal protective equipment (PPE), perceived impact of PPE on patients and workplace culture. However, this review focused only on qualitative literature, the majority of which is related to tuberculosis.

We systematically reviewed existing literature on compliance with social and behavioural protective behaviours among staff involved in healthcare, specifically during outbreaks of emerging infectious diseases and encompassing quantitative and qualitative research.

## METHODS
This rapid evidence review was carried out according to WHO guidelines[10]: the basic principles of a systematic literature review were followed, with certain aspects simplified in order to produce evidence rapidly at a time when urgent evidence synthesis is required. Searching of grey literature and quality appraisal of included studies were not carried out.

### Search strategy
The search strategy consisted of four search strings (adherence terms, protective behaviour terms, emerging infectious disease terms and healthcare worker terms). The full search strategy can be seen in online supplemental appendix I. Five databases were searched from date of inception to 4 May 2020: MEDLINE, PsycINFO, Embase, Global Health and Web of Science.

### Selection criteria
To be included, studies had to (1) contain primary data; (2) be published in peer-reviewed journals in English; (3) include participants who worked in healthcare; (4) include data on factors predicting adherence to social or behavioural infection control practices during emerging infectious disease pandemics. For the latter criterion, quantitative data needed to report statistics on factors associated with compliance, while the qualitative component of the review considered studies reporting on participants' beliefs about facilitators and barriers to compliance as well as any reported difficulties in complying with protective behaviours.

### Screening
One author (SB) ran the searches on all databases on 4 May 2020. Resulting citations were downloaded to EndNote V.X9 (Thomson Reuters, New York, USA). The same author evaluated titles for relevance, then used the inclusion criteria to screen abstracts and then full texts of remaining citations, and excluded any which were irrelevant. Any queries or uncertainties about inclusion were discussed with the wider research team. Reference lists of

all remaining papers were hand-searched for additional relevant studies.

### Data extraction and synthesis
SB extracted the following data from papers: authors, publication year, country of study, design, participants (including *n*, demographic information and profession), disease outbreak, protective behaviours measured, measures used and key results. We used thematic analysis[11] to synthesise the data and group results into themes.

### Patient and public involvement
No patients were involved in this review study.

## RESULTS
The initial search strategy yielded 1900 papers, of which 744 duplicates were removed and 1090 were excluded based on title or abstract. An additional 12 papers were found via hand-searching the reference lists of included papers. After full-text screening, 56 papers remained for inclusion. A PRISMA (Preferred Reporting Items for Systematic Reviews and Meta-Analyses) flow diagram of the process can be seen in online supplemental appendix II.

Countries represented in the literature included Canada (n=13), Saudi Arabia (n=7), Singapore (n=7), China (n=5), South Korea (n=5), USA (n=4), Netherlands (n=2), Australia (n=1), Greece (n=1), India (n=1), Iran (n=1), Taiwan (n=1), Turkey (n=1), UK (n=1) and Vietnam (n=1). A further five papers included participants from multiple countries. Papers discussed H1N1 (n=21), SARS (n=20), MERS (n=11), avian influenza (n=2) and COVID-19 (n=2). The participants represented in the literature were from a wide range of roles and departments in the healthcare profession, and a wide range of protective behaviours were considered. A full overview of study characteristics is presented in online supplemental table I.

Eight main themes were identified. These were sociodemographics and personal characteristics, occupational role, training and knowledge, work-related factors, personal protective behaviour–related factors, guidance, distress and risk perception, and attitudes and behaviours of others. Online supplemental table II provides a summary of themes and subthemes and identifies for each theme which papers showed a significant association with protective behaviours, which papers found no significant association and which papers supplemented these findings by reporting on the theme but without statistical analysis (eg, qualitative papers and papers with descriptive statistics only). Online supplemental table III summarises the evidence extracted from the literature for each theme. A number of other potential predictors of compliance were considered but only appeared in one paper each; these are presented in online supplemental table I and III but not covered in the text.

## Sociodemographics and personal characteristics

Overall, there appeared to be no significant association between age[12–18] or gender[12–14 17 19–21] and protective behaviour. Only one study[20] found that older age was significantly associated with protective behaviours, while two found that female staff were significantly more likely to comply with protective behaviours.[16 22]

One study examined nationality as a predictor[12] and found that Saudi staff were significantly more likely to comply with protective behaviours than non-Saudi staff working in the same city. Mixed findings were reported in the studies comparing behaviours across countries. Staff in Hong Kong and Singapore were more likely to comply than UK staff,[23] whereas Koh et al[24] and Wong et al[25] found that staff in Singapore versus Indonesia or Hong Kong versus Canada, respectively, were more likely to comply with some recommendations but not others (see online supplemental table III for details). A worldwide study[26] found no significant differences between countries in terms of taking protective measures.

There was no evidence of association between compliance and religion[20] or marital status.[18 19] One paper found that staff in 'high or middle' socioeconomic status were more likely to comply than those of lower socioeconomic status.[22] Four studies found that level of education was not associated with compliance,[13 18 20 21] while one found a significant association between compliance and qualification[12] (see online supplemental table III for details). One study found that H1N1 influenza vaccination was significantly associated with high compliance.[27] Having a chronic illness, being pregnant or having a pregnant spouse, elderly person or school-aged child at home were not associated with compliance, but staff with babies at home were more likely to comply with protective behaviours.[19]

## Occupational role

Many studies which examined the role as a predictor of compliance found a significant relationship[12 19 23 28–36]; however, due to the variety of different roles compared across studies, it was not possible to identify an exact pattern (see online supplemental table III for details). Five studies found no significant association between role and compliance.[13 16 17 37 38]

Length of time in role was not significantly associated with compliance in five studies,[12 16–18 30] while two studies suggested that longer experience of working in healthcare was associated with greater compliance[31 33] and one suggested that less than 10 years' experience was associated with significantly higher compliance than more experience.[36]

## Training and knowledge

There was mixed evidence on the effectiveness of outbreak-specific training and education: Taghrir et al[21] found no significant association between protective behaviours and having received education; Nour et al[17] found a non-significant increase in protective practices post-training; and Shigayeva et al[35] found that recent infection control training was a significant predictor of compliance with recommended behaviours. Jeong et al[22] found that staff who sought information about the outbreak and infection control were more likely to comply with recommended behaviours. Qualitative evidence suggested that staff felt their prior training and education were not useful in dealing with the rapidly changing nature of emerging infectious disease outbreaks.[39 40] Participants themselves believed that inadequate training was a barrier to compliance[28] and that infection control training with annual refresher courses would benefit them.[41]

Sources of knowledge about the outbreak and protective behaviours were not associated with protective behaviours in two studies,[20 21] while knowledge from textbooks and attending Continuing Medical Education activities were significantly associated with higher levels of protective practice in one study.[12] Receiving outbreak-specific training was not significantly associated with compliance, but higher outbreak-related knowledge did result in significantly higher compliance.[20] Another study examining knowledge of the outbreak itself[28] found that the majority of participants believed lack of knowledge about mode of transmission contributed to poor compliance. Knowledge of current recommendations was associated with compliance in three studies,[4 16 27] was associated with compliance in one hospital but not three other hospitals in another study,[14] and was not associated with compliance in one further study.[12] Hsu et al[42] found that a minority of participants believed that lack of education explained lack of compliance.

## Work-related factors

Compliance appeared to be higher in higher acuity settings such as emergency, intensive care or inpatient departments.[14 35 38 43 44] Wong et al[45] found that staff in high-infection districts were more likely to wear gowns, wash hands and use disinfectants but less likely to comply with quarantine measures. Two studies found no significant association between setting and compliance.[19 21]

Having contact with confirmed cases was associated with higher compliance in three studies,[14 35 43] whereas one study[19] found no association. A further study[18] found higher compliance in staff who worked directly with confirmed cases. Wong et al[45] found that SARS-exposed staff were more likely than non-SARS-exposed staff to comply with mask guidance, but less likely to quarantine themselves.

There was some evidence that high workload may be a barrier to compliance with recommended personal protective behaviours,[23 30 35 36 39 41 46] although one study also suggested that higher workload (in terms of working overtime) was associated with increased compliance in terms of giving patients appropriate infection control advice.[36]

Two studies suggested that monitoring of compliance by superiors or public health authorities encouraged compliance,[27 47] while a small minority of participants in

Hsu *et al*'s[42] study reported that better policing by infection control staff was the most important strategy for improving compliance.

Two studies suggested compliance was associated with characteristics of the patient encounter. Many of de Perio *et al*'s[14] participants reported not using recommended PPE if they did not know the patient had H1N1 or an influenza-like illness, if they did not think it was needed for the particular activity they were doing, if they only entered the patient's room for a brief time, if they did not touch the patient or if they did not come within 6 feet of the patient. Meanwhile, Shigayeva *et al*'s[35] participants were less likely to comply with recommended behaviours when providing care for patients with more severe illness (which the authors suggest may be due to the time required to don barrier equipment leading staff to put patient safety above self-protection) and if they were only observing procedures, rather than performing or assisting with them.

### Personal protective behaviour–related factors

Many studies, although mostly without statistical analysis, reported issues in performing personal protective behaviours, most notably due to lack of availability of appropriate PPE, perceived difficulty of protective behaviours, logistic issues, perceived effectiveness, perceived importance, convenience, comfort and the impact on patient care.

Participants in several studies raised concerns about lack of PPE,[14 39–42 48–53] mostly due to insufficient resources. Shortage of PPE created difficulties such as having to wear the wrong size PPE.[40 41 51] One study showed that availability of PPE was significantly associated with higher compliance,[27] while another found that staff were significantly more compliant with PPE use when eyewear and gloves were readily available at the point of care, but availability did not increase compliance of N95 respirators, surgical masks or gowns.[33] Logistic issues were also reported, such as lack of space in the hospital, making it difficult to use PPE appropriately.[51 54 55]

There may also be issues regarding the perceived difficulty of protective behaviours. Kang *et al*'s[54] participants reported difficulties with the complexity of using several PPE items together, and only 35% of van Dijk *et al*'s[53] participants thought the recommended measures were feasible.

Perceived effectiveness of PPE significantly predicted higher compliance in three studies.[27 33 34] In one qualitative study, staff expressed doubts about the quality and effectiveness of PPE.[54] Beliefs in the effectiveness of infection control procedures, as modified by previous experiences, were identified by participants as having a positive impact on compliance.[46]

Hsu *et al*[42] reported that failure to recognise the importance of hand hygiene prevented compliance, while Vinck *et al*[36] found that the main reason for not complying with the recommendation to consult with a specialist unit for centralised assessment of symptomatic patients was associated with finding it 'unnecessary'.

Participants in many qualitative studies reported not using PPE due to perceived inconvenience and its effect on their ability to do their job[14 27 39–42 54 56 57]—in particular, it appeared that participants found it too time-consuming to change in and out of protective clothing as it slowed them down substantially.

Many of Khalid *et al*'s[49] participants reported that having to use PPE was 'stressful'. Many qualitative studies found that discomfort of PPE was reported to be a barrier to compliance.[39–41 46 52 56–58] In several studies, participants reported physical harm due to PPE, including dehydration and skin peeling,[41] difficulty breathing,[40 52 58 59] sweating and dizziness,[58] headaches[40 52 60] and skin rashes.[40 52] Comfort of PPE was examined as a potential predictor of compliance in one quantitative study[33] which found that staff who reported always or often feeling comfortable wearing protective eyewear and N95 respirators were significantly more likely to wear them; however, no significant association was found in compliance between staff who reported always or often feeling short of breath, claustrophobic or dizzy when wearing protective eyewear or N95 respirators and those who rarely experienced these symptoms.

Qualitative data also revealed that many healthcare professionals believe that PPE use has an impact on patient care, making it difficult to communicate with patients due to muffled speech,[40 41 52 56 57 59] being unable to establish non-verbal cues with patients[60] and making them less 'visible' to their patients.[61] Participants in Rowlands' study[52] also reported concern that masks could be frightening for psychiatric patients, and Tan *et al*'s[40] participants reported that PPE created anxiety in some patients who assumed they were wearing it as they had been exposed to the virus. One quantitative study found that the perception that PPE use would interfere with patient care was significantly associated with poor compliance.[27]

### Guidance

Participants in many studies reported a lack of guidance on the protocols for caring for infected patients and protecting themselves.[28 30 39–41 46 50 51 53–56 58 61–65]

In particular, guidelines were often found to be too long, providing staff with 'information overload'.[30 50 63] The recommended protocols were also reported to change too frequently for staff to keep up[39 41 54–56 58 61 63 64] and staff received conflicting messages from different sources.[39 41 50 51 62–64] Other issues included inconsistent advice about what to do when patients are deemed non-infectious,[41] difficulty working out what should be prioritised[50] and external guidance with little relevance to specific locations.[39] The manner of communication of guidance could also be an issue, with Moore *et al*'s[39] participants reporting that new guidelines were sent to them by email, which they did not check before work; Rambaldini *et al*'s[64] participants reporting frustration that

information was filtered down from other sources rather than given to them first hand; and Nhan et al's[63] participants suggesting that communication about changes to protocols was too slow.

## Distress and risk perception

High levels of distress were associated with higher compliance in two studies: Chia et al[43] found that higher Impact of Events Scale scores were associated with higher use of respiratory protection, while Wong et al[45] found that staff who were highly anxious were more likely to comply with recommended protective behaviours. However, when the protective measure involved quarantine, some staff felt such high stress that they were tempted to break guidelines[47] (it is unclear from the paper what percentage of these actually did break the guidelines, if any).

Risk perception appeared to be associated with compliance. Four studies found that compliance was significantly more likely in staff who perceived the disease to be a serious risk[20 22 34 66] and another found that perceived seriousness of the outbreak was significantly associated with compliance in Hong Kong but not in Singapore or the UK.[23] Three qualitative papers also suggested that participants themselves believed staff complied with recommended behaviours when they perceived the risk to be severe.[39 47 51] However, one study[21] showed a significant negative correlation between protective behaviours and fear of infection.

## Attitudes and behaviours of others

Qualitative data revealed that many participants had observed non-compliance in colleagues or managers[27 39] which could lead to non-compliance in the participants themselves.[30 42] DiGiovanni et al's[47] participants suggested that managers had modified the recommended quarantine guidelines to allow their staff to return in response to critical staff shortages. Yassi et al's[46] participants believed that compliance of other occupational groups within the healthcare setting had an impact on compliance, while more than half of Hsu et al's[42] participants believed that senior staff 'leading by example' and complying with recommended behaviours was the single most important strategy in improving staff compliance.

Attitudes of family members were also deemed important in two qualitative studies,[39 46] with healthcare staff reporting they were encouraged to comply by anxious family members who were afraid of getting infected.

## DISCUSSION

Overall, the studies reviewed provided mixed and sometimes contradictory results. Nonetheless, some risk factors which stood out as being the most promising either for identifying specific groups at risk for poor compliance or for targeting in interventions. These included working outside of emergency or intensive care settings, not working with confirmed infection cases, lack of concern about risk of infection, lack of monitoring by superiors,

observed non-compliance of colleagues, lack of PPE, perceived difficulty using PPE, perceived lack of effectiveness or lack of importance of PPE, perceived inconvenience and discomfort of PPE, perceived negative impact of PPE on patient care, lack of infection control guidance, and inconsistent or unclear guidance. Organisations faced with nosocomial transmission would be wise, as part of their mitigation efforts, to look to these areas in an effort to help staff adhere to protective behaviours.

There was little evidence for an association between compliance with protective behaviours and sociodemographic or personal characteristics. There was some evidence that compliance levels differ across countries: it is not clear whether this is due to between-country differences in the communication of guidance, different risk perceptions due to different countries' media coverage of the outbreak, different levels of training received or cultural differences in the participants themselves.

This review suggests that compliance may differ between different roles and different settings. Targeted interventions for specific occupational groups with different levels of patient contact, hierarchies and cultures may be helpful. While it was unclear from the varied data which occupational groups might need more attention, it seems that those not in emergency departments or intensive care units and not working directly with infected patients could benefit from additional focus.

We found little evidence that training and education significantly improved compliance. This initially seems surprising as previous trials of training courses to improve knowledge of infection control practices and encourage compliance with them have suggested positive results.[67–69] However, taken together with the evidence that compliance differs between occupational groups, it may be that staff in different roles require different levels and methods of training. We also found mixed evidence of the impact of knowledge about infection control on compliance with the recommended measures: this may be associated with what is actually covered in training. For example, as it appears that the perceived importance of PPE is related to compliance, it may be that training that covers the 'why' rather than the 'how' of PPE is particularly useful. However, training itself should not be viewed as a panacea. Many of the factors identified by our review, and discussed below, are not amenable to better training.

A small amount of evidence suggested that observed non-compliance by others—including colleagues—could affect healthcare workers' own levels of compliance. This is likely to be particularly problematic where managers and supervisors fail to 'lead by example' and ensure that they comply with the recommended policies and procedures. 'Role modelling' by superiors could be useful, with supervisors setting the standards for infection control practices and reinforcing them.

The availability, ease of use, perceived effectiveness, convenience, comfort and impact on patients of PPE may be other key factors which need to be addressed. This review suggests fairly negative views of PPE, with

participants reporting lack of appropriate resources, the inconvenience of having to change in and out of PPE repeatedly, physical discomfort and a negative impact on their ability to communicate with patients. It is useful to know about these concerns about PPE as this can be a target to change.

Reviewed studies suggest difficulties understanding and keeping up with rapidly changing recommendations. This may be unavoidable during an outbreak of an emerging infectious disease, as new evidence becomes available. Staff should therefore be prepared for the likelihood of changes, both to help maintain trust and to ensure that they monitor for updates.

There was some evidence that higher levels of anxiety and risk perception were associated with higher compliance. This may suggest that, while it is important not to create unhealthy anxiety, desensitisation to risk or genuine reductions in risk may lead to a reduction in PPE use. Indeed, it has been reported that staff in industrial settings experience complacency in terms of protective behaviours, as a result of habituation (repeated exposures which do not result in anything bad, lessening fearful responding) and low levels of positive reinforcement for safe behaviour.[70] Particular care may be needed to maintain PPE use as risk changes, or is perceived to change, over the course of an outbreak.

## Limitations

We are aware that many papers (particularly on SARS and MERS) have been published in other (predominantly Asian) languages, but due to the rapid nature of this review we limited inclusion to English-language papers only. Future reviews should consider translating and analysing the many relevant foreign-language papers. Also, due to the rapid nature of the review, quality appraisal of individual papers was not carried out as this is not always deemed necessary when urgent evidence synthesis is required.[71] Our searches were carried out in May 2020 in order to provide an early look at the factors associated with nosocomial transmission at the start of the COVID-19 pandemic, and we are aware that new relevant studies have been published since this date. It would be useful for future reviews to consider the newer literature on COVID-19 and compare the results with our own. We also note that how adherence changes over the course of a long outbreak that is characterised by periods of high and low transmission is not clear—the studies we included did not address this, as most focused on relatively short outbreaks and the few COVID-19 papers that were included were from the early days of the pandemic. It is plausible that the factors driving behaviour change as time goes on, and more research is needed.

## CONCLUSION

Interventions that may be helpful for improving healthcare workers' compliance include role-specific or setting-specific training, emphasising the importance of protective behaviours and the risk of infection if behaviours are not performed, monitoring of staff behaviours by supervisors and positive reinforcement for correct behaviours, managerial staff leading by example, training focused on the importance and effectiveness of PPE, and better communication of guidelines. The results of this review also suggest that the following interventions would likely *not* be useful: training aimed at specific age or gender groups, training focused on increasing knowledge about the outbreak itself or training on how to use PPE without also emphasising *why* it is necessary.

**Contributors** GJR conceived the work. SB conducted the literature search and screened search results for inclusion, and GJR, NG and SW resolved any queries about inclusion. All authors contributed to the qualitative results synthesis, prepared the first draft of the manuscript and approved the final manuscript.

**Funding** The research was funded by the National Institute for Health Research Health Protection Research Unit (NIHR HPRU) in Emergency Preparedness and Response at King's College London in partnership with Public Health England (PHE), in collaboration with the University of East Anglia (funding reference: NIHR200890).

**Disclaimer** The views expressed are those of the author(s) and not necessarily those of the NHS, the NIHR, the Department of Health and Social Care, or Public Health England. The funder had no involvement in the study design, collection, analysis or interpretation of data, writing the report or decision to submit for publication.

**Competing interests** None declared.

**Patient consent for publication** Not required.

**Provenance and peer review** Not commissioned; externally peer reviewed.

**Data availability statement** No data are available. No original data were generated for this study.

**ORCID iDs**
Samantha K Brooks http://orcid.org/0000-0003-3884-3583
Simon Wessely http://orcid.org/0000-0002-6743-9929

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
