## [Reviewer comments · BMJ Open]

ARTICLE DETAILS

TITLE (PROVISIONAL)	Factors affecting healthcare workers' compliance with social and behavioural infection control measures during emerging infectious disease outbreaks: Rapid evidence review.
AUTHORS	BROOKS, SAMANTHA; Greenberg, N; Wessely, Simon; Rubin, GJ

VERSION 1 – REVIEW

REVIEWER	Musu, Mario Università degli Studi di Cagliari
REVIEW RETURNED	14-Mar-2021

GENERAL COMMENTS	The work rigorously addresses many of the problems of adherence to standards and recommendations that many clinicians empirically observe in clinical practice. It is therefore useful to help caregivers to have behaviors more consistent with best clinical practices. However, I believe that focused and controlled clinical studies would be needed to have useful data to reflect on what to do and how to correct daily mistakes.
---

REVIEWER	Thomas, Hannah Telethon Kids Institute, Population Health
REVIEW RETURNED	24-Mar-2021

GENERAL COMMENTS	This review is well written, in clear and concise language. It synthesizes the information generated by a large number of studies to provide recommendations on potential targets to increase healthcare workers' compliance with infection control measures. In the context of the evolving COVID-19 pandemic, these recommendations are topical and required. The authors acknowledge the limitations of the study, including the selection of only English language publications for inclusion. Further, I note the literature search was conducted in May 2020. At this time minimal manuscripts were published addressing this question in the context of COVID-19, however a short search reveals additional papers published more recently which may further contribute to understandings in this area. As such, this review is a timely early look at an important issue, and the field will benefit from further extensive reviews in the future.
--

REVIEWER	Donker, Tjibbe University Clinic Freiburg, Institute for Infection Prevention and Hospital Epidemiology
REVIEW RETURNED	16-Apr-2021

GENERAL COMMENTS	Brooks and colleagues performed a systematic review on the what factors influence HCW compliance with infection control measures. The manuscript is well written, the methods are easy to follow, and the results are meticulously tracked. I Have very few comments on the paper. The paper selection was done completely by a single author (SKB). Was there a system in place to ensure that any "borderline" papers, matching the inclusion criteria only just (or not), were re-evaluated by the other authors? The author contribution section seems to elude to this, but it might be good to mention this in the methods section. As a minor point maybe, the conclusion isn't very much to the point, in my opinion. A bit of rewriting may help the readability here. Please try and focus on what this review could find in potential recommendations, instead of focussing on the inconsistencies it found, which are not really surprising, I'd say. Therefore, try and focus on what can be recommended first. I have no further comments.
--

VERSION 1 – AUTHOR RESPONSE

Reviewer: 1

Comment: The work rigorously addresses many of the problems of adherence to standards and recommendations that many clinicians empirically observe in clinical practice. It is therefore useful to help caregivers to have behaviors more consistent with best clinical practices.

However, I believe that focused and controlled clinical studies would be needed to have useful data to reflect on what to do and how to correct daily mistakes.

Response: We would like to thank the reviewer for their feedback and we agree that focused, controlled studies would be a useful addition to the literature.

Reviewer: 2

Comment: The authors acknowledge the limitations of the study, including the selection of only English language publications for inclusion. Further, I note the literature search was conducted in May 2020. At this time minimal manuscripts were published addressing this question in the context of COVID-19, however a short search reveals additional papers published more recently which may further contribute to understandings in this area. As such, this review is a timely early look at an important issue, and the field will benefit from further extensive reviews in the future.

Response: We agree with this comment and have added the following to the limitations section: "Our searches were carried out in May 2020 in order to provide an early look at the factors associated with nosocomial transmission at the start of the COVID-19 pandemic, and we are aware that new relevant studies have been published since this date. It would be useful for future reviews to consider the newer literature on COVID-19 and compare the results to our own."

Reviewer: 3

Comment: The paper selection was done completely by a single author (SKB). Was there a system in place to ensure that any "borderline" papers, matching the inclusion criteria only just (or not), were re-evaluated by the other authors? The author contribution section seems to elude to this, but it might be good to mention this in the methods section.

Response: We have added the following to the Methods section: "Any queries or uncertainties about inclusion were discussed with the wider research team."

Comment: As a minor point maybe, the conclusion isn't very much to the point, in my opinion. A bit of rewriting may help the readability here. Please try and focus on what this review could find in potential recommendations, instead of focussing on the inconsistencies it found, which are not really surprising, I'd say. Therefore, try and focus on what can be recommended first.

Response: We have removed the sentence on inconsistencies from the conclusion.

VERSION 2 – REVIEW

REVIEWER	Thomas, Hannah Telethon Kids Institute, Population Health
REVIEW RETURNED	09-Jun-2021

GENERAL COMMENTS	The authors have adequately addressed all reviewer and editor comments. Congratulations on this contribution to the literature.
--

REVIEWER	Donker, Tjibbe University Clinic Freiburg, Institute for Infection Prevention and Hospital Epidemiology
REVIEW RETURNED	15-Jun-2021

GENERAL COMMENTS	The authors have addressed my previous questions and comments. However, I would like to add that emphasising that the paper focuses on the factors affecting compliance during the *early* stages of the pandemic may help the readers appreciate what has been done. This requires no vast amount of rewriting and certainly no new literature search. It would, though, improve the appeal of the paper, as the relatively long time between literature search and (potential) publication is very obvious to the reader. Furthermore, it is not unthinkable that compliance with control measures changed during the pandemic period. later studies may therefore show different results. I would therefore want to urge the authors to consider carefully reading through their manuscript, and rewriting parts of it where needed. This despite the authors properly addressing my previous comments.
---

VERSION 2 – AUTHOR RESPONSE

Comments to the Author: The authors have addressed my previous questions and comments.

However, I would like to add that emphasising that the paper focuses on the factors affecting compliance during the *early* stages of the pandemic may help the readers appreciate what has been done. This requires no vast amount of rewriting and certainly no new literature search. It would, though, improve the appeal of the paper, as the relatively long time between literature search and (potential) publication is very obvious to the reader.

Furthermore, it is not unthinkable that compliance with control measures changed during the pandemic period. later studies may therefore show different results.

I would therefore want to urge the authors to consider carefully reading through their manuscript, and rewriting parts of it where needed. This despite the authors properly addressing my previous comments.

RESPONSE: We have added the following to the limitations section of the paper:

"We also note that how adherence changes over the course of a long outbreak that is characterised by periods of high and low transmission is not clear – the studies we included did not address this, as most focused on relatively short outbreaks and the few COVID-19 papers that were included were from the early days of the pandemic. It is plausible that the factors driving behaviour change as time goes on, and more research is needed."